SOFTWARE

# NeKo: A tool for automatic network construction from prior knowledge

**Marco Ruscone** [iD][1,2,3,4*], **Eirini Tsirvouli** [iD][5], **Andrea Checcoli** [iD][1,2,3,6], **Denes Turei**[7], **Emmanuel Barillot**[1,2,3], **Julio Saez-Rodriguez**[7,8], **Loredana Martignetti**[1,2,3], **Åsmund Flobak** [iD][5], **Laurence Calzone**[1,2,3*]

**1** Institut Curie, Université PSL, Paris, France, **2** INSERM, U900, Paris, France, **3** Mines ParisTech, Université PSL, Paris, France, **4** Barcelona Supercomputing Center (BSC), Barcelona, Spain, **5** Department of Clinical and Molecular Medicine, Norwegian University of Science and Technology, Trondheim, Norway, **6** Centre de Recherche des Cordeliers, Sorbonne Université, Paris, France, **7** Faculty of Medicine and Heidelberg University Hospital, Institute for Computational Biomedicine, Heidelberg University, Heidelberg, Germany, **8** European Bioinformatics Institute, European Molecular Biology Laboratory, Hinxton, United Kingdom

\* marco.ruscone@bsc.es (MR); laurence.calzone@curie.fr (LC)

**Data availability statement:** The source code and data used to produce the results and

## Abstract

Biological networks provide a structured framework for analyzing the dynamic interplay and interactions between molecular entities, facilitating deeper insights into cellular functions and biological processes. Network construction often requires extensive manual curation based on scientific literature and public databases, a time-consuming and laborious task. To address this challenge, we introduce NeKo, a Python package to automate the construction of biological networks by integrating and prioritizing molecular interactions from various databases. NeKo allows users to provide their molecules of interest (e.g., genes, proteins or phosphosites), select interaction resources and apply flexible strategies to build networks based on prior knowledge. Users can filter interactions by various criteria, such as direct or indirect links and signed or unsigned interactions, to tailor the network to their needs and downstream analysis. We demonstrate some of NeKo's capabilities in two use cases: first we construct a network based on transcriptomics from medulloblastoma; in the second, we model drug synergies. NeKo streamlines the network-building process, making it more accessible and efficient for researchers.

## Author summary

Biological networks are a powerful tool for understanding complex diseases, providing insight into molecular interactions and regulatory mechanisms. However, constructing these networks is a challenging and time-consuming task, particularly when relying on manual curation from existing literature. To reduce the burden of analyzing thousands of scientific articles, researchers can extract interactions from biological databases. While

analyses presented in this manuscript are available on the following GitHub repositories: https://github.com/sysbio-curie/Neko; https://github.com/sysbio-curie/NeKo_Supp_Mat.

**Funding:** AC was financed by ModICeD project financially supported by ITMO Cancer of Aviesan (2020). LM is part of ModPhosphoNet, a project financially supported by ITMO Cancer of Aviesan (2021-2030). LC and MR are partially supported by Certainty project which is part of the European Union's Horizon Europe research and innovation program under grant agreement n101136379. DT was supported by the HPC/Exascale Centre of Excellence for Personalised Medicine in Europe [PerMedCoE; European Union Horizon 2020 program, grant no. 951773] and by the German Federal Ministry of Education and Research [Bundesministerium für Bildung und Forschung BMBF; grant no. 031L0181B] funds granted to JSR. EB and LC are part of PRAIRIE 3IA Institute, funded by the French government under the management of ANR as part of the Investissements d'avenir program (ANR-19-P3IA-0001). JSR reports funding from GSK, Pfizer, and Sanofi and fees/honoraria from Travere Therapeutics, Stadapharm, Astex, Owkin, Pfizer, Moderna, and Grunenthal.

**Competing interests:** The authors have declared that no competing interests exist.

many online tools exist for building networks from databases, they often lack flexibility. What is missing is a versatile framework that allows users to construct networks using multiple approaches and from diverse sources, including offline datasets. In this article, we introduce NeKo, a Python package designed to automate network construction by integrating interactions from various sources while offering users control over the building process. We demonstrate its utility by applying NeKo to Medulloblastoma data and by comparing a manually curated network of gastric cancer (Cascade) with one generated automatically by NeKo. Our results highlight the potential of NeKo in streamlining network inference, bridging the gap between manual curation and automated network generation.

## 1. Introduction

Many biological processes, such as intracellular signaling, can be represented and studied as networks of interacting molecular entities. Networks can provide structured representations of the dynamic interplay between various biological levels and can facilitate a detailed appreciation of the underlying regulatory mechanisms governing cellular functions and decision-making.

Networks can be tailored to depict biological processes at varying levels of detail, ranging from broad and abstract overviews to highly detailed representations of molecular processes. Two approaches can be applied for the construction of networks: a *data-driven* and a *knowledge-driven* approach. Data-driven approaches are based on high-throughput experimental data, such as gene expression or phosphoproteomics, to infer regulatory interaction networks that underpin cellular functions. On the other side, knowledge-driven network construction involves the comprehensive review of scientific literature and extensive manual curation of information from databases that collect known interactions, such as SIGNOR [1] or Reactome [2]. While data-driven approaches adapt well to various data types and are particularly useful for large-scale analyses, they remain highly sensitive to data quality and can lead to potential overfitting or inaccuracies due to noise and high-dimensional data complexity [3,4]. On the other hand, a knowledge-driven approach provides more comprehensive networks as they gather information from many different sources, but they might be biased or incomplete because they are based on the current extent of biological knowledge.

To account for those limitations, data-driven and knowledge-driven approaches are frequently combined during network construction [5–7]. To achieve such a combination, several tools and strategies have been developed to integrate omics data and prior knowledge, including SignalingProfiler [8], CARNIVAL [9] and Augusta [10]. However, even in hybrid approaches, manual curation is still required to a great extent and remains a cumbersome and laborious task. During the manual curation step, a lot of different databases are typically queried, each collecting interactions based on different criteria, standards, biological focus, and approaches. To facilitate the harmonization of those sources of knowledge, efforts such as OmniPath [11] and Pathway Commons [12,13] have been developed to collect and harmonize interactions from various resources, making the integrated knowledge readily available for the user. However, this process could be further automatized by implementing strategies similar to how a curator builds a problem-specific network from a general compendium of interactions. Here, we present NeKo, a Python tool to automatically construct biological networks by employing a series of flexible strategies to extract, group, and merge molecular interactions from various databases. Given a list of molecular entities of interest (called *seeds*)(e.g., a list of differentially expressed genes or proteins) and a pre-defined source of interactions

(e.g., public database or proprietary), NeKo enables users to select various strategies to connect the seeds. NeKo is able to consider or ignore the direction and causality of the interactions. The final network can be used as it is for further analysis, or amended by the user (e.g., removing nodes, edges or paths) before being exported in the various formats provided by NeKo. In the following sections, we describe in detail the network-building strategies offered by the tool and provide two use cases to showcase potential workflows to construct and use NeKo-derived networks. NeKo is a free software available at https://github.com/sysbio-curie/Neko.

## 2. Design and implementation

### 2.1. Requirements

NeKo is a Python package designed to be used with Jupyter Notebooks, offering a streamlined and intuitive approach for collecting and summarizing prior knowledge based on the input provided.

To install NeKo, we provide a step-by-step procedure in the Supplementary Material in S1 Text and at the following link: https://sysbio-curie.github.io/Neko/. To avoid issues with the different operating systems and versions of required packages, we strongly suggest installing and running the tool within a conda or Poetry environment.

### 2.2. Workflow

The input of NeKo can be a list of molecular entities (genes, proteins), the *seeds*. NeKo then uses knowledge reported in databases to construct a new network, or to expand an existing network provided by the user. NeKo searches in the specified databases for the seeds and establishes links between them, using various connection strategies. Moreover, NeKo provides the option to annotate the network and suggests how the resulting network connects to different ontology terms based on Gene Ontology (Sect 2.3.4). The result of the workflow is a network that can be further analyzed with other tools such as Cytoscape [14] for topological analysis, or a Boolean model in *BNET* format (Sect 2.3.5) for dynamical modeling. An example of a typical NeKo workflow is presented in Fig 1.

First, the user selects the database of interest and the seeds. By default, NeKo derives interactions from OmniPath [11], which allows extractions of interactions from multiple sources, and NeKo can also be configured to use other public or proprietary databases. We expect that the reader will access the OmniPath manuscript for additional information on sources covered. We also added simple direct clients to SIGNOR and PhosphoSitePlus because of their frequent updates and high significance in the field. For a specification of these databases, see Sect 2.3.1. The seeds refer to the entities to be connected by NeKo and can be provided using gene symbols.

As a next step, the user chooses one or more connecting strategies to create the network (*see* Sect 2.3.3 for more details). NeKo also provides methods for the editing of the networks it created (e.g., removing nodes, interactions, or paths). Furthermore, NeKo is able to include in the networks biological processes or phenotypes, by directly accessing the Gene Ontology database (*see* Sect 2.3.4). The final network can be visualized and exported in various formats, supporting a broad variety of downstream applications (*see* Sect 2.3.5).

### 2.3. Package structure

The NeKo package consists of six Python classes, each one with a specific function:

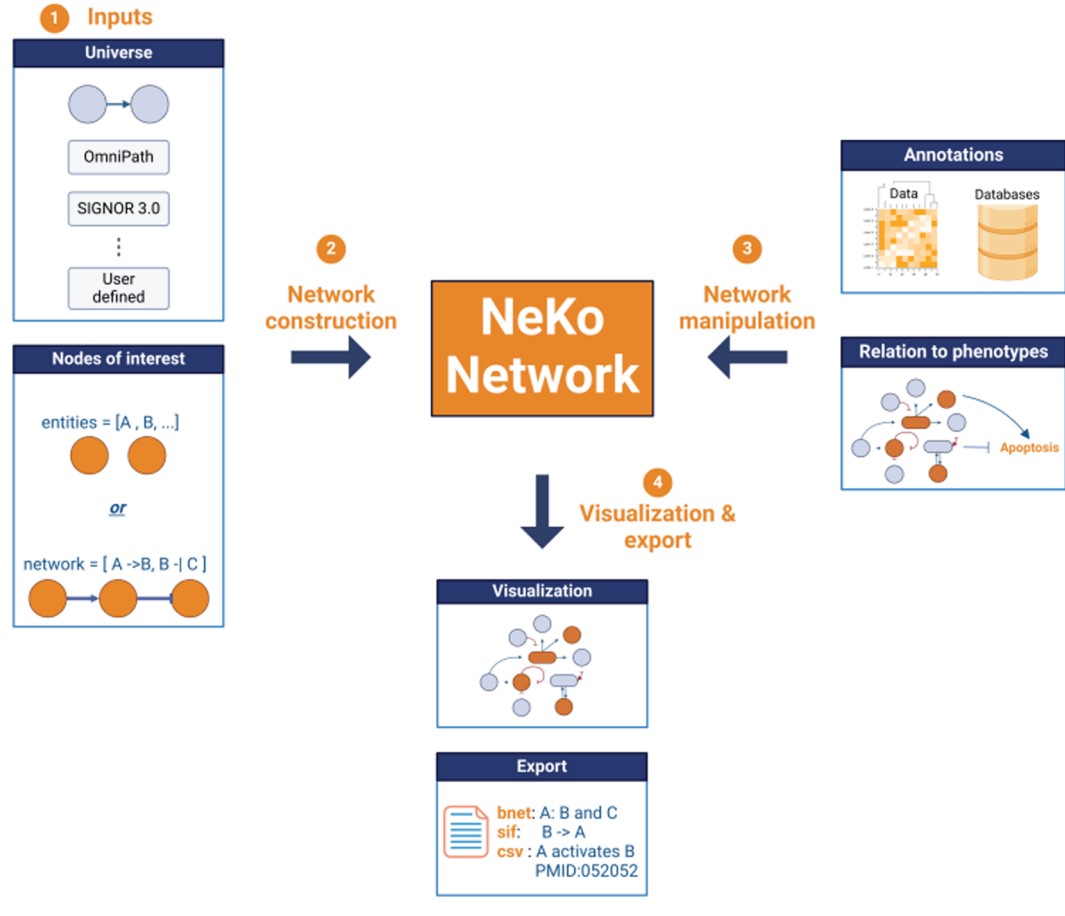

**Fig 1. Example of a typical workflow to construct a network using NeKo.** First, the universe of interactions and the seed nodes of interest are defined. Then, NeKo will connect the seed nodes based on the user-defined algorithm. The resulting network can be manipulated by additing annotations or connected to phenotypes using Gene Ontology terms. Lastly, the final network can be visualized and exported in various formats. Created in BioRender. Ruscone, M. (2025) https://BioRender.com/z98d148

- `Network`: hosts the structure and the functionalities of a network built with NeKo.
- `Universe`: provides methods to build a database from external sources, store it, and make it accessible to the other classes.
- `Connections`: implements basic algorithms for path-finding (such as Breadth First Search [15] and Depth First Search [16]), as well as more advanced algorithms for identifying any paths (see the function `find_upstream_cascades`).
- `Ontology`: fetches nodes associated to a given ontology term and connects them to the network.
- `Exports`: exports the network in different formats, such as *BNET* and *SIF* (Simple Interaction Format).
- `NetworkVisualizer`: visualizes the network and exports the figures to *pdf* format.

**2.3.1. The network class.** The `Network` class is represented by two `pandas` data frames: `Network.nodes` and `Network.edges` Fig 2A. The `Network.nodes` data frame contains the Gene Symbols and UniProt identifiers of the entities in the network. The `Network.edges` data frame stores the edges of the network, each defined by its source

### Network nodes

|   | Genesymbol | Uniprot |
|---|---|---|
| 0 | SRC | P12931 |
| 1 | NOTCH1 | P46531 |
| 2 | PTK2 | Q05397 |
| 3 | CDH1 | P12830 |
| 4 | CDH2 | P19022 |
| … | … | … |

### Network edges

A

|   | source | target | Type | Effect | References |
|---|---|---|---|---|---|
| 0 | P12931 | Q05397 | post_translational | stimulation | Adhesome:10085298;Adhesome:10592173;Adhesome:1… |
| 1 | P12830 | P12931 | post_translational | stimulation | ACSN:16039586;ACSN:16099633;ACSN:17143292;ACSN… |
| 2 | P12931 | P19022 | post_translational | inhibition | ACSN:15782139;ACSN:16371504;ACSN:16492141;ACSN… |
| 3 | P12931 | Q14289 | post_translational | stimulation | Adhesome:10329689;Adhesome:10521452;Adhesome:1… |
| 4 | P46531 | P19022 | transcriptional | stimulation | CollecTRI:16618740; CollecTRI:16618740 |
| … | … | … | … | … | … |

### Universe DataFrame

B

|   | source | target | is_directed | is_stimulation | is_inhibition | form_complex | consensus_direction | consensus_stimulation | consensus_inhibition | curation_effort | references | sources |
|---|---|---|---|---|---|---|---|---|---|---|---|---|
| 0 | A0A024RAD5 | SIGNOR-C535 | True | False | False | True | False | False | False | miannu | 31831667 | SIGNOR-272062 |
| 1 | A0A0B4J2F0 | P18848 | False | False | True | False | False | False | False | miannu | 31653868 | SIGNOR-261041 |
| 2 | A0A0B4J2F0 | P35638 | False | False | True | False | False | False | False | miannu | 31653868 | SIGNOR-261043 |
| 3 | A0A0B4J2F0 | SIGNOR-PH2 | False | False | True | False | False | False | False | miannu | 31653868 | SIGNOR-261042 |
| 4 | A0AVT1 | SIGNOR-C496 | True | False | False | True | False | False | False | miannu | 24816100 | SIGNOR-270835 |
| … | … | … | … | … | … | … | … | … | … | … | … | … |

**Fig 2. Structure of the node dataframe (upper, left panel), edge dataframe (upper, right panel) and Universe dataframe (bottom panel) used in the NeKo package.**

and `target` nodes, and characterized by an `effect`, which can be 'stimulation', 'inhibition', 'bimodal', 'form_complex', or 'undefined', depending on the available information of each resource and the user inputs. Additionally, `Network.edges` includes information about the edge type and literature references if these are available in the database.

To create a `Network` object two arguments are required: a list of seed nodes and a data frame representing the database adapted to the NeKo format. The seeds and any known interactions can also be provided as a network to be expanded in a *SIF* file format.

The `Network` class also includes methods for adding and removing nodes, edges, or entire paths. It also offers various strategies that users can invoke to find and merge paths within the network. Users can customize the path collection strategies, for instance by filtering for signed paths (more details in the Supplementary material (S1 Text), Additional Information, Sect S3 "Link between Python functions and NeKo strategies").

**2.3.2. The Universe class.** The `Universe` class stores the prior knowledge database in pandas data frames Fig 2B. By default, the complete OmniPath interactions database is used.

The `Universe` data frame has a column layout similar to the OmniPath interaction data frames. By default, it contains the following columns: 'source', 'target', 'is_directed', 'is_stimulation', 'is_inhibition', 'form_complex', 'consensus_direction', 'consensus_stimulation', 'consensus_inhibition', 'references', and 'sources', as shown in Fig 2. Of these columns, only the 'source' and 'target' are mandatory (defining to an unsigned network by default). The `Universe.add_database` method fills the missing columns with empty values of `False`. Additionally, NeKo provides specific functions to import the SIGNOR database [1] and the PhosphositePlus (PSP) [17], including the PSP regulatory sites dataset about the effects of phosphorylation sites on protein activities. These two databases were directly integrated into NeKo to support a higher update frequency and access to additional details compared to what

is currently available in OmniPath. We chose to include support for these databases because of their high quality.

Additional information on how to build a database for NeKo can be found in the notebook "Build network using user-defined resources" (https://sysbio-curie.github.io/Neko/notebooks/2_add_resources.html).

**2.3.3. The connections class.** NeKo offers different approaches, or strategies, to aggregate paths to build a network, starting from the seeds provided by the user. These strategies are based on well-known algorithms for path finding, such as Breadth-First Search (BFS) and Depth-First Search (DFS) algorithms.

The `Connections` class in NeKo implements these algorithms and provides a set of methods to explore and analyze networks efficiently. This class utilizes a pandas data frame as its underlying data structure, representing the edges from the prior knowledge database with 'source' and 'target' columns. To optimize performance, the class preprocesses the resource data into dictionary-based data structures, allowing for quick lookups of neighboring nodes.

Key features of the `Connections` class include methods for finding target and source neighbors, performing BFS and DFS to find one or multiple paths between selected nodes. The class also incorporates more advanced functionalities, such as finding upstream cascades and identifying minimal sets of regulators that cover specified target genes (with the function `find_cascades`). These methods enable an efficient exploration of complex relationships and regulatory patterns within biological networks.

**2.3.4. The ontology class.** This class is designed to map a subset of nodes in the network to a certain phenotype. It allows the user to specify a GO accession code and the relative ontology term, to fetch the relative marker nodes from GeneOntology (https://geneontology.org/). In the same class, we also provide an option to map tissue specific expression to the network nodes, using data from the Human Protein Atlas [18], accessed via the OmniPath Annotations database.

**2.3.5. The exports class.** The `Export` class outputs the networks in formats that are readily usable by popular downstream analysis tools. The SIF output is supported by Cytoscape [14] and many other software.

NeKo is also able to generate Boolean models from the networks it created. These can be exported in BNET format. A Boolean model is a mathematical representation of a biological system where each variable (node) can have one of two states: ON (1) or OFF (0), and the state of each node is determined by Boolean functions that link its input nodes with logical connectors, AND, OR and NOT. To create these models, NeKo automatically generates Boolean equations associated with each node in the network. The process for generating these equations is heuristic: all activators of a node are chained by OR gates, as are all inhibitors; the groups of activators and inhibitors are then separated by an AND NOT gate.

If the network encompasses nodes that form a complex (especially when fetching interactions from SIGNOR), the Boolean equation of the complex will have its members chained by AND gates. Furthermore, if the network contains bimodal interactions (those that can both activate and inhibit), an ensemble of models is created by permuting across all possible combinations of activation and inhibition for all bimodal interactions in the network. These generic rules can provide a starting point for further refinement based on expert knowledge. Tools such as BoNeSiS [19] can be used to refine these equations based on experimental data or additional biological knowledge.

The `Export` class thus serves as a bridge between NeKo's network construction capabilities and the wider ecosystem of network analysis and modeling tools.

**2.3.6. The NetworkVisualizer.** The class `NetworkVisualizer` converts NeKo's networks to Graphviz objects (https://graphviz.org/), which support visualization. The network can be visualized directly in the Jupyter Notebook or exported to PDF. Moreover, it provides functions to customize the visualization, such as coloring nodes of interest, or to highlight the inputs and outputs of a specific node.

## 2.4. Connection strategies

Building on the classes presented previously, the `Network` class implements advanced strategies for network construction and expansion, using the functions in the `Connection` class. These strategies are illustrated in Fig 3 and described in detail below:

- **Reciprocal Pathway Extender (RPE) strategy:** It ensures comprehensive connectivity between node pairs by searching for shortest paths and integrating neighboring interactions. This algorithm initiates by checking if each pair of nodes is connected in both directions using the BFS or DFS algorithm. If a connection in one or both directions is missing, the algorithm searches for the shortest paths to establish these connections, incorporating all intermediate nodes and edges into the network. It then integrates direct neighboring interactions of newly added nodes before moving to the next pair.

- **Iterative Neighbor Expansion (INE) strategy:** It builds networks through progressive incorporation of neighboring nodes. This algorithm begins by exploring each node's immediate neighbors and integrating them into the network. If the remaining nodes are disconnected, the network is expanded by adding the neighboring nodes of the newly added nodes from the previous iteration. The process is repeated until all nodes are connected or until the user-defined number of iterations is reached. After completing the designated loops, the network is refined by removing any disconnected nodes or non-regulated nodes (i.e., nodes without any sources).

- **Regulatory Cascade Explorer (RCE) strategy:** It identifies and ranks upstream regulators of designated output nodes, allowing for deep exploration of regulatory networks. After identifying all regulators for each node, the regulators are then ranked based on the total number of output nodes they regulate. Importantly, the ranking is flexible and adjusts to the actual degree of influence: the more output nodes a regulator interacts with, the higher it ranks (see Fig 3 for an example). By modifying the "depth" argument of the function, the algorithm looks further back to the regulators of these initial regulators, applying the same flexible ranking method at each level, based on their influence over their immediate successors. This strategy is particularly useful for identifying and prioritizing common transcription factors of gene targets that correspond to seeds in the network.

- **Module Connection Mapper (MCM) strategy:** It establishes connections between predefined groups of nodes, utilizing the RPE algorithm to refine pathways between groups. It comprehensively iterates through all possible pairings between the groups to list existing pathways. Each pathway's intermediate nodes are aggregated into a separate group, Group C. Subsequently, the Reciprocal Pathway Extender (RPE) algorithm is applied exclusively to this intermediary group to refine and extend the network pathways between the original groups. The MCM algorithm allows for three modes of connections: IN, OUT, and ALL. For the IN and OUT modes, only directed interactions from the "source" towards the "target" group of nodes are included.

- **Phenotype Integration and Network Connectivity (PINC) strategy:** It integrates phenotype-specific genes into existing networks using the Gene Ontology (GO)

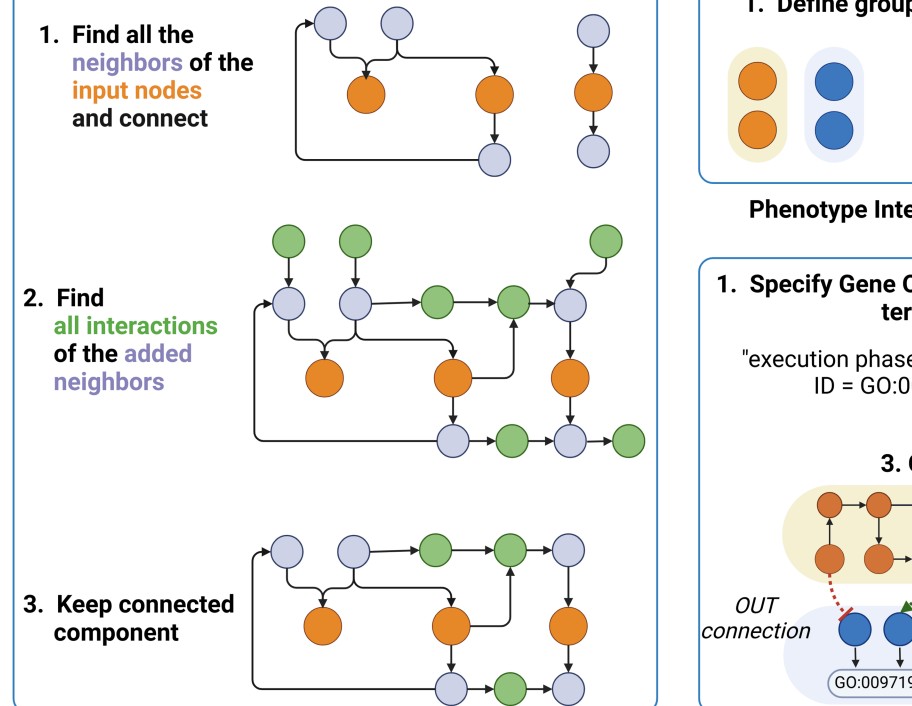

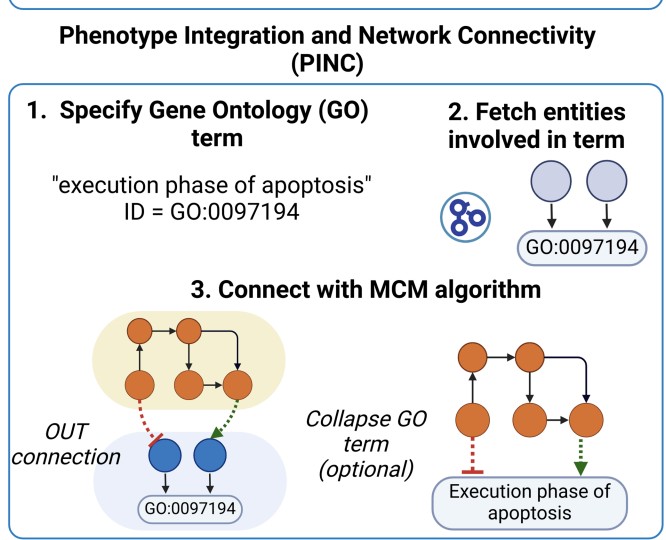

**Fig 3. Overview of the connection strategies provided by NeKo.** Orange nodes denote the user-input nodes. Created in BioRender. Ruscone, M. (2025)
https://BioRender.com/c93h960.

annotations and the MCM algorithm. Starting with a phenotype of interest, users input a corresponding GO accession ID and its name, allowing the algorithm to fetch all associated genes from the GO database. These genes are then organized into a new group, which is integrated into the user's existing network using the afore-described MCM

algorithm. Finally, to simplify and enhance the network's interpretability, the user can decide to merge the connected phenotype-related genes into a single composite node. This node represents the phenotype and maintains all interactions inherited from the individual genes, offering a clear and consolidated view of the interactions between specific phenotypes and the existing network.

## 2.5. NeKo strategies for building a network

The network construction strategies available in NeKo have been designed to work in both a complementary and synergistic fashion. NeKo users can apply one or multiple strategies to build and refine biological networks originating from a set of biological entities. As an example, a typical use of NeKo begins with creating a baseline network by applying a single iteration of the Iterative Neighbor Expansion (INE) strategy on a set of seed nodes, corresponding to genes or proteins of interest. This set of nodes could be a list of genes provided by experts, the list of top hits resulting from a differential analysis (either proteomics- or transcriptomics-based), or a signature derived from a statistical analysis. The resulting network may connect the nodes, either directly or indirectly, while an unconnected network suggests missing components.

In the case of an unconnected network, several strategies can be proposed, independently from each other or consecutively.

If the nodes are not connected, there may exist intermediate entities in the database that would connect them. The search for intermediate nodes can be done with the Reciprocal Pathway Extender (RPE) strategy with the choice to include one, two, or more intermediates and connect them to the current network.

The Regulatory Cascade Explorer (RCE) Sect 2.3.3 can facilitate the retrieval of entities that regulate more than one node, and provide complementary information to the network by determining common regulators for a given set of nodes.

The Module Connection Mapper (MCM) strategy aims at connecting two separate groups of nodes, for example, a set of receptors with a set of downstream transcription factors. This strategy does not handle all nodes together, but as two groups, creating paths from one to the other.

The Phenotype Integration and Network Connectivity (PINC) strategy connects phenotypes to the existing network by replacing all genes that belong to a phenotype with a single phenotype node. This method enhances biological interpretability and provids a more comprehensive view of the biological system.

Alternatively, modelers can base their reasoning on sets of nodes derived from gene lists associated with specific ontology terms, i.e., cell cycle, or apoptosis. By combining multiple ontology terms, the corresponding nodes can be merged into a network (whose components can be connected following any of the available strategies presented above), thereby highlighting the relationships between these ontology terms and other nodes provided by the user.

Overall, the network construction strategies available in NeKo provide a flexible toolkit to handle different types of biological data and types of inferred networks (e.g., protein-protein interaction networks, regulatory networks, TF networks, etc.).

The methods presented here complete almost instantly when building small networks ($\leq 50$ nodes), while above 300 nodes the process may take up to two minutes (more details in the documentation, in the notebook "Re-creating famous pathways from SIGNOR and WIKIPATHWAYS using NeKo").

## 3. Results

We showcase NeKo's application to real-life questions in two case studies on cancer datasets.

### 3.1. Use Case 1 - Medulloblastoma subgroup network

Medulloblastoma, the most common pediatric brain tumor in childhood, can be classified into four distinct subgroups, each characterized by unique genetic signatures, clinical features, and outcomes [20]. The wingless (WNT) and sonic hedgehog (SHH) subgroups have well-defined profiles with abnormalities in developmental pathways. Group 3 and Group 4 are less well-defined but with Group 3 characterized by MYC amplification and Group 4 by CDK6 and SRC overexpression [21].

Previous analyses of patient stratification based on omics data [21] have shown that Groups 3 and 4 share some similarities but the underlying mechanisms are different. In Group 3, deregulations at the transcription level are reported, while in Group 4, alterations of post-transcriptional events are observed involving proteins like SRC or ERBB4.

Signatures based on transcriptome and DNA methylome profiles are adopted in clinical practice, but we lack a clear understanding of their causal relationship to the disease. We propose, with NeKo, to construct a network from these signatures and expect to see the emergence of new signaling pathways when connecting the genes of the signatures that could bring additional and insightful information.

Given that the assembled network topology is heavily influenced by the choice of the database, we opted to fetch causal interactions from SIGNOR (directed and signed interactions) and protein-protein interactions from HuRI (undirected and unsigned interactions), employing identical workflows and algorithms for both databases.

The network construction process starts with collecting genes/proteins of interest from major publications that focus on medulloblastoma [22]. The seed genes specific to each subgroup are reported in Table 1. It is important to note that not all the selected genes were present in the chosen databases. When querying the databases, NeKo ignores the genes of interest that are missing from the Universe.

In this use case, we applied the RPE strategy, using the depth-first search algorithm, to connect the seed genes while minimizing the amount of intermediate nodes. We selected a

**Table 1. Seed genes for each subgroup.**

| WNT | | SHH | | Group 3 | Group 4 | |
|---|---|---|---|---|---|---|
| ADGRB3 | IRX2 | BCOR | PRKAR1A | ATM | BRCA2 | MYCN |
| ALX4 | LEF1 | BRCA2 | PTCH1 | CRX | CDK6 | OTX2 |
| APC | LHX8 | CEBPA | PTEN | HLX | EOMES | POU2F1 |
| ARID1A | MAF | CREBBP | RARB | IRX6 | FOXP2 | PRDM6 |
| ATM | MSX2 | DDX3X | SOX13 | LHX9 | GFI1 | SIX1 |
| CSNK2B | OSR2 | FBXW7 | SOX2 | MYC | GFI1B | SIX6 |
| CTNNB1 | PAX3 | GLI2 | SUFU | PTEN | KDM6A | TAL1 |
| DDX3X | PIK3CA | GSE1 | TBX18 | SMARCA4 | KMT2C | TBR1 |
| DLX3 | PITX1 | KMT2C | TCF4 | | KMT2D | ZIC1 |
| EMX2 | PRRX1 | KMT2D | TERT | | LHX1 | ZMYM3 |
| HOXC4 | RUNX2 | NFATC1 | TP53 | | LHX2 | |
| HOXC5 | SYNCRIP | PBX1 | | | LMX1A | |
| HOXD8 | POU6F2 | OTX2 | | | MED12 | |

Seed genes used to generate the networks for each subgroup. Those genes were selected from two major publications focused on medulloblastoma analysis, Northcott et al. [22] and Lin et al. [26].

maximum path length of 3, which was increased to 4 for disconnected nodes. If the resulting network contained disconnected nodes, the RPE strategy was de novo applied using breadth-first search.

The networks were then exported in SIF format for visualization and topological analysis using Cytoscape.

An additional step was implemented for the networks built from SIGNOR to incorporate complex and protein family nodes, introducing all complex components and connecting them to the rest of the network.

We performed a GO enrichment analysis using the Cytoscape plug-in ClueGO [23] to assess whether the network constructed for each subgroup exhibited characteristic features (Fig 4) and compared it to the GO analysis before the network construction. The GO analysis was performed using the same settings for all subgroups (intermediate terms and p-value $\leq 0.05$). The results show that after the network construction, mechanisms coherent with the known biology appear for each group, whereas the usual GO enrichment analysis with the seed genes falls short of providing a mechanistic explanation (S6 Fig). Indeed, the Wnt signaling pathway is strongly enriched in the NeKo network of Wnt subgroup confirming that Wnt/Wingless pathway activation characterize this distinct molecular subgroup of medulloblastomas [24]. Additionally, we find cell cycle-related terms significantly enriched in the network of Group 3, which is known for its accelerated proliferation driven by MYC activation [25]. Finally, the network of Group 4 shows an enrichment of kinase-receptor terms, consistent with the fact that this subgroup is particularly characterized by post-transcriptional regulatory events [21]. More details about the results are provided in Supplementary Material (S4, S5, and S6 Figs).

## 3.2. Use Case 2 - DrugLogics pipeline

The DrugLogics pipeline [27] is designed to predict drug synergy using a genetic algorithm to calibrate logical models to data. As input, the pipeline receives a network file in SIF format that contains signed and directed interactions. A manually curated logical model referred to as 'Cascade', describing the signaling interactions around seven drug targets in gastric cancer, previously demonstrated high predictive accuracy for identifying synergistic drug combinations [28]. To assess whether a NeKo-derived topology could achieve comparable performance with the manually curated topology, we constructed a NeKo topology using as an input the same seven drug targets as in the initial publication.

NeKo successfully connected all the input nodes resulting in a network of 57 nodes and 140 edges. Compared to the manually curated network by [28], which consisted of 77 nodes and 217 edges, the NeKo-derived network was relatively smaller, but more densely connected as shown by its higher average clustering coefficient (i.e., 0.4 for NeKo-network and 0.07 for the manually curated one). A more detailed topological comparison of the two networks is presented in the Supplementary Information (S7 Fig).

To assess whether the network generated by NeKo successfully captured the signaling events underlying drug responses, we evaluated its ability to identify experimentally validated synergistic drug combinations, as also presented in [28]. The network obtained via Neko correctly classified 3 of the 4 synergies (i.e., true positives), compared to the initial model which classified them all correctly.

For a more comprehensive comparison of their overall performance, the ROC (Receiver Operating Characteristic) curves of the two models are provided (Fig 5). The ROC-AUC (Receiver Operating Characteristic Area Under the Curve) values provide a metric that assesses the model's ability to distinguish between synergies and non-synergies. A value of

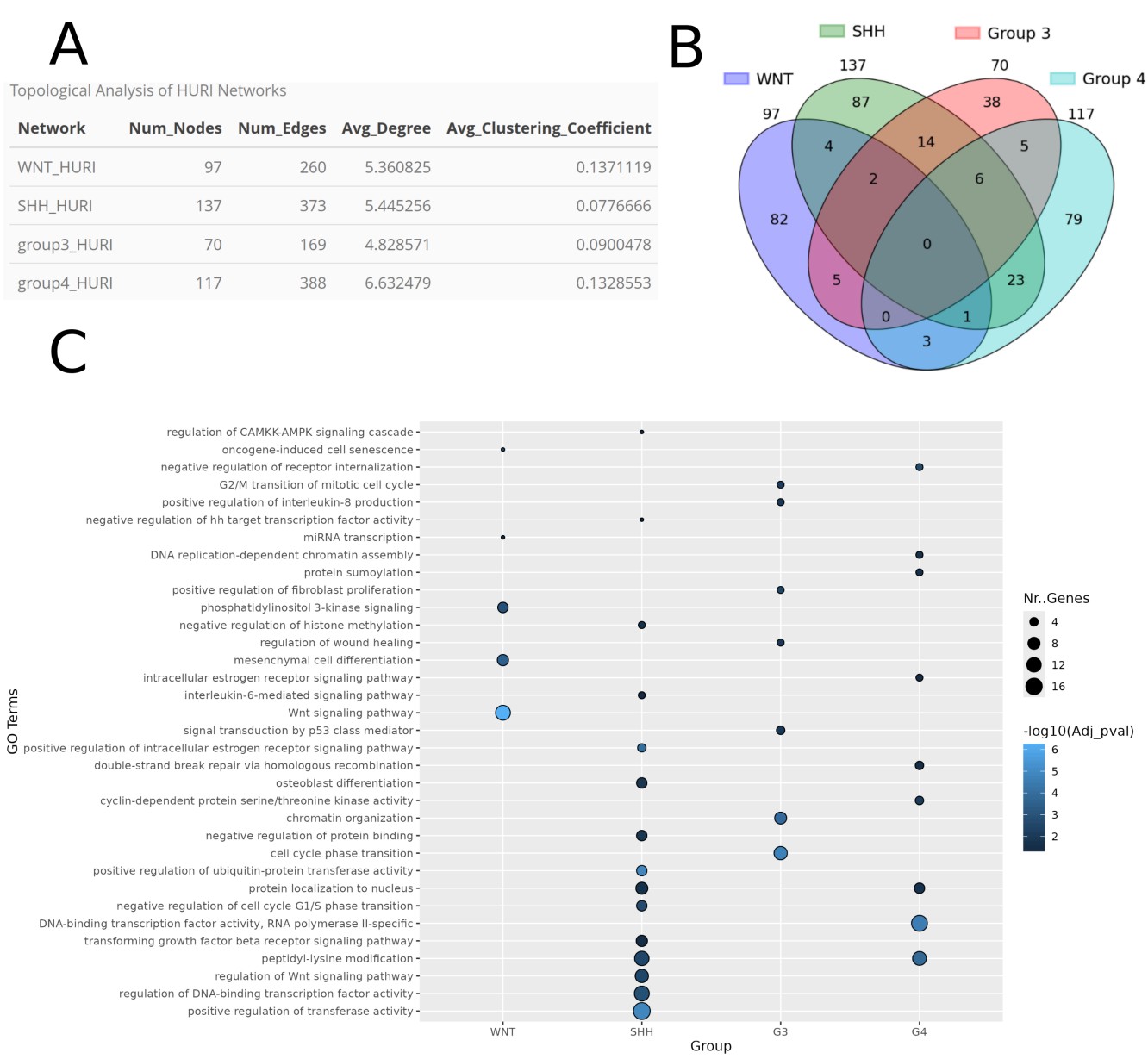

**Fig 4. Topological and functional comparison of the NeKo networks created for the medulloblastoma subgroups using the HURI databases.** A) Comparison of the main topological network properties between the four subgroups. B) Venn diagram displaying the common nodes among each subgroup. C) Functional comparison of the four medulloblastoma subgroup networks generated from the HURI database, using the ClueGO Cytoscape plug-in. Only leading terms with an adjusted p-value ≤ 0.05 are presented.

1 denotes a perfect model that correctly classifies all combinations. A value of 0.5 implies that the model is random, with no predictive ability. In the case of the NeKo model, an AUC value of 0.65 is reported, highlighting that the model has a significant predictive ability, even with no user intervention to the resulting topology. This also indicates that the NeKo model includes biologically relevant interactions, that correctly capture the necessary events triggered in drug responses.

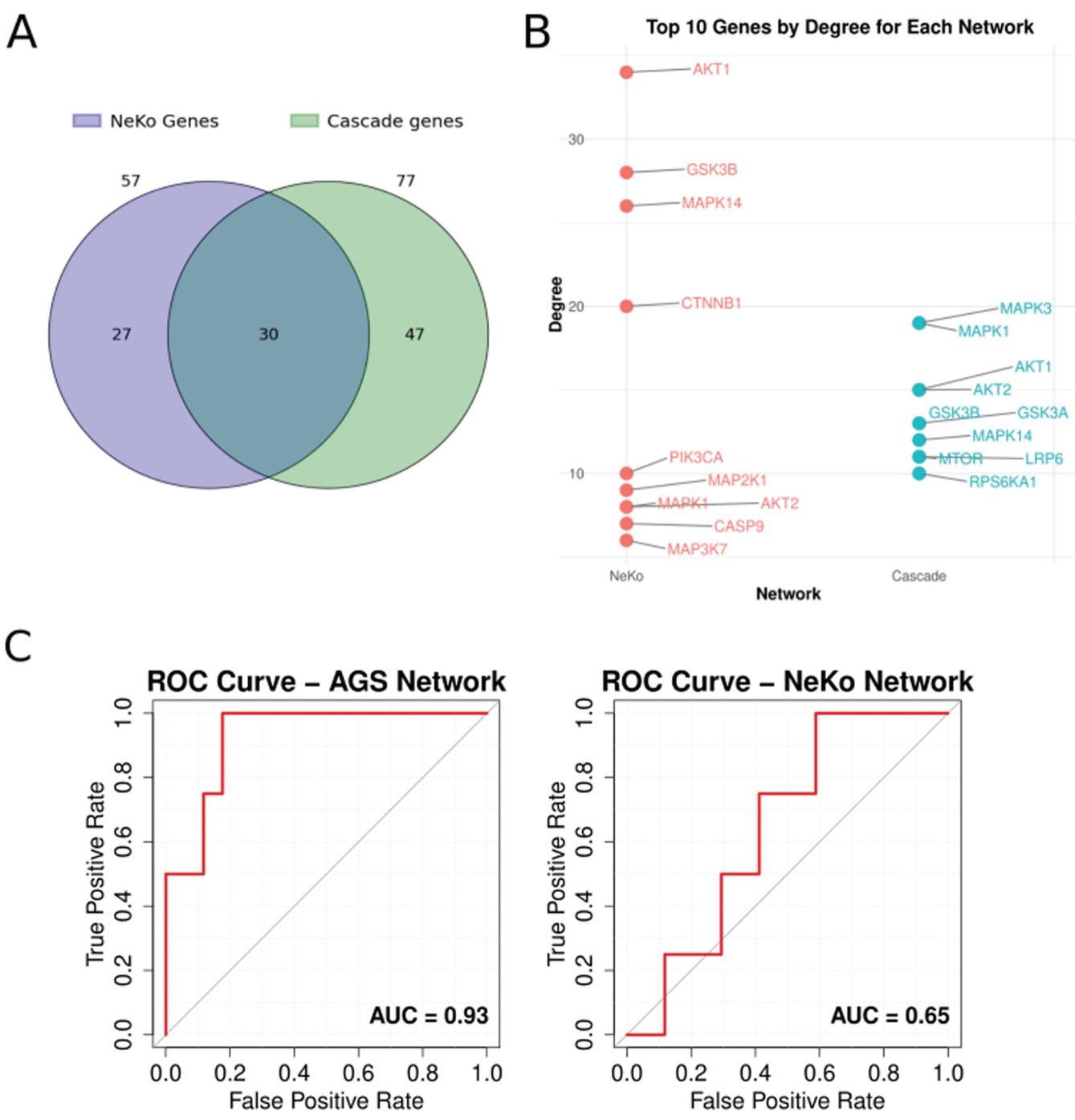

**Fig 5. Comparison of the results obtained combining NeKo and Druglogics pipeline with the original publication from [28].** A) Venn diagram showing the overlap between the genes included in the two models, NeKo-inferred network vs. Cascade network. B) The top 10 nodes with the highest degree for each network. C) The ROC curves to assess the ability of the two models to correctly predict synergistic drug combinations (*left panel*: Cascade model, *right panel*: NeKo model).

However, the manually-curated model, which contains validated processes reported in the literature, performs relatively better (Fig 5). The performance indicates that the network from NeKo is a useful starting point for further manual curation, saving substantial time compared to manual curating.

## 4. Discussion

NeKo offers users a versatile and efficient tool for building networks from prior knowledge databases. Its ability to automate the integration of molecular interactions and tailor network building to specific research facilitates the otherwise labor-intensive and time-consuming process of manual curation.

NeKo's adaptability was demonstrated by two use cases related to cancer applications. The first use case on medulloblastoma showed the possibility of interpreting molecular signatures with networks and gaining some insights into the processes that are deregulated in each subgroup, which may not have been obvious with the sole signatures. The next step could be the translation of this network into a mathematical model to study dynamically the four subgroups. NeKo facilitates the process by providing a first draft of the network and an initial list of logical equations with standard rules. With the second use case, we compared two models that were developed for the same purpose of studying drug synergies and with the same initial list of genes, one built with manual curation and the other one with automatic inference by NeKo. We showed that NeKo can create biologically relevant models that can be used to identify synergistic drug combinations.

NeKo can be used as a first step for further analyses, such as topological studies, integration of omics data into these networks [29], or visualization of data onto the networks (with the SIF output), and modeling purposes (with BNET output). The type of downstream analyses that users intend to perform will also affect the network construction strategy. Depending on whether the aim is to conduct network analysis or build a predictive model, the choices regarding the number of genes, the selected database, and the level of detail in inferred interactions might differ significantly.

Despite these strengths, the effectiveness of NeKo is inherently limited by several factors and should be considered as a building block of a pipeline that would lead to further refinement and analyses. First, the scope of the analysis is confined by the selection of input entities (genes, proteins, phosphosites, etc.). The seed nodes significantly influence the resulting network, and any omission or misidentification of key molecular players can lead to incomplete or biased networks. Additionally, the choice of databases plays a crucial role in shaping the network. Different databases have varying levels of comprehensiveness, curation quality, and focus, which can impact the accuracy, completeness, and relevance of the constructed networks. Integration within a larger ecosystem of network biology, such as the Network Commons (https://github.com/saezlab/networkcommons), would allow synergizing NeKo with other components, including algorithms for network contextualization from omics data [7], complementary visualizations, and downstream benchmark.

Another limitation resides in the Boolean models that can be built with NeKo. The heuristic approach used to shape the Boolean equations that characterize each node of the network does not lead to any significant attractor in most cases. This is because NeKo does not rely on data to support or constrain specific network states, meaning that the attractors reached when simulating the Boolean model often do not have biological relevance. The main issue arises from the large number of edges introduced into the network, which generates excessive noise and reduces the interpretability of the model. Future improvements could involve refining the edge selection process to reduce noise and improve the meaningfulness of attractors.

Finally, the graphic library used to visualize the network, Graphviz, struggles with performance and readability when dealing with a high number of nodes in Jupyter notebooks. We plan in future updates to support multiple libraries for graph visualization, such as Igraph or NetworkX, to provide more efficient and interactive alternatives.

In summary, NeKo represents a significant step forward in the automation and flexibility of biological network construction. Its ability to integrate diverse interaction data and provide customizable network-building strategies makes it a valuable tool for users aiming to explore complex biological systems.

## 5. Availability and future directions

NeKo is currently available in PyPi (refer to Supplementary Material (S1 Text) for detailed installation instructions). The package documentation, including the full API reference, is automatically generated using Sphinx [30]. This documentation includes several Jupyter Notebooks, each of which offers step-by-step guidance for building a NeKo network, providing practical examples to help users become familiar with the workflow. The use cases demonstrated in this manuscript are publicly accessible on GitHub (https://github.com/sysbio-curie/NeKo_Supp_Mat), with additional insights and analyses provided in the Supplementary Information (S1 Text). Regular updates will ensure continued support and expansion of NeKo's features. Future versions will introduce new functionalities, including methods for the automated refinement and curation of NeKo networks based on user-specified criteria, such as data integration or prior knowledge input.

Additional export options will also be implemented to generate output files compatible with other modeling formalisms, such as Ordinary Differential Equation (ODE) models. Finally, we plan to refactor NeKo's codebase to facilitate contributions from the community and enhance its accessibility. Since NeKo works best when integrated with other modeling and analysis tools, we will ensure that its structure remains modular and compatible with broader computational pipelines, including the aforementioned Network Commons.

## Supporting information

**S1 Text. This document contains additional details and analysis that complement the work presented in the main text.** The code and figures used for both the main article and the supplementary materials can be found at the following github page: https://github.com/sysbio-curie/NeKo_Supp_Mat.
(PDF)

## Author contributions

**Conceptualization:** Marco Ruscone, Julio Saez-Rodriguez, Laurence Calzone.

**Methodology:** Marco Ruscone, Eirini Tsirvouli, Andrea Checcoli, Denes Turei.

**Software:** Marco Ruscone, Eirini Tsirvouli, Andrea Checcoli, Denes Turei.

**Validation:** Marco Ruscone, Eirini Tsirvouli, Loredana Martignetti, Åsmund Flobak, Laurence Calzone.

**Writing – original draft:** Marco Ruscone, Eirini Tsirvouli, Andrea Checcoli, Laurence Calzone.

**Writing – review & editing:** Marco Ruscone, Eirini Tsirvouli, Andrea Checcoli, Denes Turei, Emmanuel Barillot, Julio Saez-Rodriguez, Loredana Martignetti, Åsmund Flobak, Laurence Calzone.

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
