## [Decision Letter · Decision Letter 0]

6 Jan 2025

PCOMPBIOL-D-24-01942

NeKo: a tool for automatic network construction from prior knowledge

PLOS Computational Biology

Dear Dr. Ruscone,

Thank you for submitting your manuscript to PLOS Computational Biology. After careful consideration, we feel that it has merit but does not fully meet PLOS Computational Biology's publication criteria as it currently stands. Therefore, we invite you to submit a revised version of the manuscript that addresses the points raised during the review process.

Please submit your revised manuscript within 60 days Mar 08 2025 11:59PM. If you will need more time than this to complete your revisions, please reply to this message or contact the journal office at ploscompbiol@plos.org. Please include the following items when submitting your revised manuscript:

We look forward to receiving your revised manuscript.

Kind regards,

Alberto J M Martin, Ph.D.

Academic Editor

PLOS Computational Biology

Mark Alber

Section Editor

PLOS Computational Biology

**Additional Editor Comments:**

Please do address issues raised by reviewers, specially those regarding escalation on larger datasets, limitations and validation of the tool

**Journal Requirements:**

4) Please ensure that all Figure files have corresponding citations and legends within the manuscript. Currently, Figure 2 in your submission file inventory does not have an in-text citation. If the figure is no longer to be included as part of the submission, please remove it from the file inventory.

5) We have noticed that you have uploaded Supporting Information files, but you have not included a list of legends. Please add a full list of legends for your Supporting Information files after the references list.

Potential Copyright Issues:

- The following Figure contains a logo or branding: Figure 1. We are not permitted to publish this under our CC-BY 4.0 license, even with permission. We ask that you please remove or replace it.

7) Please amend your detailed Financial Disclosure statement. This is published with the article. It must therefore be completed in full sentences and contain the exact wording you wish to be published.

**Reviewers' comments:**

Reviewer's Responses to Questions

**Comments to the Authors:**

Reviewer #1: Please see the attachment in the online system.

Reviewer #2: The manuscript "NeKo: a tool for automatic network construction from prior knowledge" provides valuable insights into biological networks in the filed of transcriptomics. The authors have done excellent work in developing a Python package for construction of biological networks by integrating molecular interactions from various databases. The study aims to develop a Python package for construction of molecular network and also model drug synergies. The tool is available in github. However, the following issues need to be addressed before publication.

1) Which databases are integrated for genes and drugs to be addressed more specifically.

2) Comparison with similar packages and its advances to be highlighted.

3) Limitations of NeKo are to listed.

Reviewer #3: Review of “NeKo: a tool for automatic network construction from prior knowledge”

The manuscript introduces NeKo, a Python-based tool designed to automate the construction of biological networks by integrating molecular interactions from multiple prior knowledge databases. The authors aim to reduce the time-consuming and labor-intensive manual curation traditionally required in knowledge-driven network construction while providing flexibility. The manuscript highlights two use cases: constructing a network for medulloblastoma subgroup-specific signatures and modeling drug synergies using logical models.

NeKo represents a significant step forward in streamlining network construction and demonstrates the potential for broad application in systems biology. However, several major and minor issues need to be addressed to enhance the manuscript's clarity, rigor, and utility.

Major Comments

1. The authors provide a solid section on strategy selection, including descriptions of available algorithms and their applications. However, as strategy selection is a key challenge for usability, this section could be broadened to include more detailed guidance, examples, or decision trees to assist users in selecting the optimal strategy for their specific research questions. Clearer recommendations would improve accessibility for non-expert users and reduce the learning curve.

2. NeKo’s model for predicting drug synergies performed relatively poorly compared to the manually curated model. The manuscript should discuss whether the chosen strategy was optimal for this task and provide insights into how strategy selection influences performance. Was the poor result due to inherent limitations in the method or suboptimal strategy alignment?

3. While the manuscript discusses NeKo’s integration with databases like OmniPath and SIGNOR, a systematic evaluation of how database choice impacts network quality is important. This is particularly substantial given NeKo's reliance on external resources, and the lack of such benchmarking may limit confidence in its results.

4. While NeKo is tested on small-to-medium networks, its performance on larger datasets (>1000 nodes) is not discussed. Computational benchmarks assessing runtime and memory usage across varying network sizes would be helpful, especially for researchers handling large omics datasets.

5. Comparing NeKo’s functionality and outputs with widely used alternatives, such as STRING or Ingenuity Pathway Analysis (IPA), would provide context for its capabilities. Highlighting NeKo's strengths (e.g., flexibility, export formats) and limitations relative to these tools would better position it within the field of network construction.

Minor Comments

- The manually curated network's AUC value is not reported in Figure 5, making it difficult to assess the performance gap between NeKo-derived and curated networks. This metric should be included for a complete comparison.

- On page 4, the text mentions "activation" edges, while Figure 2 refers to "stimulation."

- On page 9, a citation appears as "(REF)." This needs correction.

- On page 10, "Addittional" contains a typographical error.

**Have the authors made all data and (if applicable) computational code underlying the findings in their manuscript fully available?**

Reviewer #1: Yes

Reviewer #2: Yes

Reviewer #3: Yes

PLOS authors have the option to publish the peer review history of their article (what does this mean?). If published, this will include your full peer review and any attached files.

Reviewer #1: No

Reviewer #2: **Yes: **Dr. Dicky John Davis G

Reviewer #3: **Yes: **Ábel Fóthi

**Figure resubmission:**
---

## [Decision Letter · Decision Letter 1]

1 May 2025

PCOMPBIOL-D-24-01942R1

NeKo: a tool for automatic network construction from prior knowledge

PLOS Computational Biology

Dear Dr. Ruscone,

Thank you for submitting your manuscript to PLOS Computational Biology. After careful consideration, we feel that it has merit but does not fully meet PLOS Computational Biology's publication criteria as it currently stands. Therefore, we invite you to submit a revised version of the manuscript that addresses the points raised during the review process.

Please submit your revised manuscript within 60 days Jul 01 2025 11:59PM. If you will need more time than this to complete your revisions, please reply to this message or contact the journal office at ploscompbiol@plos.org. Please include the following items when submitting your revised manuscript:

We look forward to receiving your revised manuscript.

Kind regards,

Alberto J M Martin, Ph.D.

Academic Editor

PLOS Computational Biology

Mark Alber

Section Editor

PLOS Computational Biology

**Journal Requirements:**

Please amend your detailed Financial Disclosure statement. This is published with the article. It must therefore be completed in full sentences and contain the exact wording you wish to be published.Please ensure that the funders and grant numbers match between the Financial Disclosure field and the Funding Information tab in your submission form. Note that the funders must be provided in the same order in both places as well.

2) State what role the funders took in the study. If the funders had no role in your study, please state: "The funders had no role in study design, data collection and analysis, decision to publish, or preparation of the manuscript.".

**Reviewers' comments:**

**Figure resubmission:**
---

## [Decision Letter · Decision Letter 2]

7 Jul 2025

Dear Dr. Ruscone,

We are pleased to inform you that your manuscript 'NeKo: a tool for automatic network construction from prior knowledge' has been provisionally accepted for publication in PLOS Computational Biology.

Best regards,

Mark Alber, Ph.D.

Section Editor

PLOS Computational Biology

Mark Alber

Section Editor

PLOS Computational Biology

Reviewer's Responses to Questions

**Comments to the Authors:**

Reviewer #3: Thank you for your answer.

**Have the authors made all data and (if applicable) computational code underlying the findings in their manuscript fully available?**

Reviewer #3: None

PLOS authors have the option to publish the peer review history of their article (what does this mean?). If published, this will include your full peer review and any attached files.

Reviewer #3: **Yes: **Ábel Fóthi

---

## [Editor Report · Acceptance letter]

PCOMPBIOL-D-24-01942R2

NeKo: a tool for automatic network construction from prior knowledge

Dear Dr Ruscone,

I am pleased to inform you that your manuscript has been formally accepted for publication in PLOS Computational Biology. Your manuscript is now with our production department and you will be notified of the publication date in due course.

With kind regards,

Judit Kozma
